

# Acceleration-induced transport of quantum vortices in joined atomtronic circuits

Andrii Chaika[1⋆], Artem O. Oliinyk[1], Ihor V. Yatsuta[2], Nick P. Proukakis[3†], Mark Edwards[4‡], Alexander I. Yakimenko[1,5∘] and Thomas Bland[3,6§]

**1** Department of Physics, Taras Shevchenko National University of Kyiv, 64/13, Volodymyrska Street, Kyiv 01601, Ukraine
**2** Department of Condensed Matter Physics, Weizmann Institute of Science, Rehovot 7610001, Israel
**3** School of Mathematics, Statistics and Physics, Newcastle University, Newcastle upon Tyne, NE1 7RU, United Kingdom
**4** Department of Biochemistry, Chemistry, and Physics, Georgia Southern University, Statesboro, Georgia 30460-8031, USA
**5** Dipartimento di Fisica e Astronomia Galileo Galilei, Universitá di Padova, and INFN, Sezione di Padova, Via Marzolo 8, 35131 Padova, Italy
**6** Universität Innsbruck, Fakultät für Mathematik, Informatik und Physik, Institut für Experimentalphysik, 6020 Innsbruck, Austria

⋆ andriy31415@gmail.com , † nikolaos.proukakis@ncl.ac.uk , ‡ edwards@georgiasouthern.edu , ∘ alexander.yakimenko@gmail.com , § thomas.bland1991@gmail.com

## Abstract

Persistent currents–inviscid quantized flow around an atomic circuit–are a crucial building block of atomtronic devices. We investigate how acceleration influences the transfer of persistent currents between two density-connected, ring-shaped atomic Bose-Einstein condensates, joined by a tunable weak link that controls system topology. We find that the acceleration of this system modifies both the density and phase dynamics between the rings, leading to a bias in the periodic vortex oscillations studied in T. Bland *et al.*, Phys. Rev. Research 4, 043171 (2022). Accounting for dissipation suppressing such vortex oscillations, the acceleration facilitates a unilateral vortex transfer to the leading ring. We analyze how this transfer depends on the weak-link amplitude, the initial persistent current configuration, and the acceleration strength and direction. Characterization of the sensitivity to these parameters paves the way for a new platform for acceleration measurements, for which we outline a proof-of-concept ultracold double-ring accelerometer.

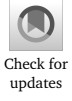

## 1  Introduction

The study of quantum vortices in Bose-Einstein condensates (BECs) has become a cornerstone of contemporary quantum fluid dynamics, offering profound insights into macroscopic quantum phenomena. Quantum vortices, characterized by quantized circulation, are fundamental to understanding superfluidity and have far-reaching implications in areas ranging from condensed matter physics to astrophysics. The precise control and manipulation of these vortices are essential for developing next-generation quantum technologies, including atomtronic circuits [1–3], novel quantum sensors [4,5] and quantum information processing systems [6].

Unlike quantum gases in conventional harmonic traps, ring-traps can support multiply quantized vortices known as persistent currents [7–19] (for a recent review see Ref. [20]), superfluid flow that can circulate indefinitely without dissipation. The ability to engineer and control such currents through external perturbations like weak links and potential barriers is critical for creating robust atomtronic devices, which are the quantum analogs of electronic circuits. While numerous proposals suggest using persistent currents for precision rotation measurements [18, 21, 22] and gyroscopy [23], existing ultracold atomic accelerometers and gravimeters [24–31] do not yet utilize persistent currents as their core mechanism.

This work examines the dynamics of persistent current transfer between rings in an accelerating double-ring system, using a time-dependent weak link to explore the feasibility of this setup for precision acceleration measurements. The introduction of acceleration to an otherwise static double-ring system is shown to induce a notable bias in the vortex transition dynamics. When combined with sufficiently strong dissipation–which is known to completely suppress persistent current oscillations in a static system [32]–acceleration alone can, by altering the phase and density of the matter distribution in the double-ring, trigger a persistent current transfer to the leading ring. As such, we demonstrate that the engineered vortex transport through linked toroidal condensates with a tunable weak link can be exploited for precise local acceleration measurements, taking advantage of the sensitivity of such transfer to barrier amplitude, persistent current distribution and acceleration.

## 2 Persistent current oscillations in the accelerating double-ring system

### 2.1 Double-ring geometry and numerical model

We consider an atomic quantum gas confined in a side-by-side double-ring geometry, inspired by Refs. [32, 33]. The gas is confined within a potential

$$V_{\text{ext}}(\mathbf{r}, t) = V_{\text{d}}(\mathbf{r}) + V_{\text{b}}(\mathbf{r}, t), \tag{1}$$

that is composed of two parts: the static double-ring potential

$$V_{\text{d}}(\mathbf{r}) = \frac{M\omega_r^2}{2} \min\left((|\mathbf{r} + R\mathbf{n}| - R)^2, (|\mathbf{r} - R\mathbf{n}| - R)^2\right), \tag{2}$$

and time-dependent barrier

$$V_{\text{b}}(\mathbf{r}, t) = V_0(t)\Theta(R - |\mathbf{r} \cdot \mathbf{n}|)e^{-[\mathbf{r} \times \mathbf{n}]^2/2l_b^2}. \tag{3}$$

Here, $\mathbf{r} = (x, y)$, $\mathbf{n} = (\cos\theta, \sin\theta)$, and $\theta$ is the angle between the direction of acceleration and the line which connects the centers of the rings. At time $t = \Delta t$, the barrier amplitude $V_0(t)$ is linearly increased from zero, such that it acts as a gate between the two rings, modifying the system topology and facilitating the transfer of the quantum vortex as discussed in Ref. [32].

A constant acceleration in the system can be implemented either by explicitly moving the potential in the laboratory frame, or by moving to an accelerating frame. As discussed later, the subsequent addition of a simple phenomenological model for thermal dissipation points us to the latter option as a more realistic model for our present purposes. In our chosen accelerating frame, our system is stationary, with acceleration dynamics appearing as a correction to the potential of the form $M\mathbf{a} \cdot \mathbf{r}$, where $\mathbf{a} = (a_x, a_y)$.

With that in mind, the zero-temperature dynamics of the condensate wave function in an accelerating frame are well described by the quasi-two-dimensional Gross–Pitaevskii Equation (GPE)

$$i\hbar\frac{\partial\psi}{\partial t} = \left(-\frac{\hbar^2}{2M}\nabla^2 + V_{\text{ext}} + g_{\text{2D}}|\psi|^2 + M\mathbf{a} \cdot \mathbf{r} - \mu_{\text{2D}}\right)\psi. \tag{4}$$

Here, $g_{\text{2D}} = \sqrt{2\pi}\hbar^2 a_s/Ml_z$ is the effective two-dimensional two-body local interaction coupling strength, $a_s$ is the background $s$-wave scattering length, $M$ is the atomic mass, and $l_z = \sqrt{\hbar/M\omega_z}$ is the harmonic oscillator length in $z$. As an example realistic system, we consider parameters inspired from the experiment of Ref. [34], but with atom number $N = 10^6$. This corresponds to $^{23}$Na atoms with $a_s = 2.75$ nm, and we also choose $\omega_r = 2\pi \times 134$ Hz, $\omega_z = 2\pi \times 550$ Hz, $R = 22.6$ $\mu$m, and a potential width of $l_b = 3.45$ $\mu$m. This gives a chemical potential of $\mu_{\text{2D}}/\hbar = 2000$ Hz. The density is normalized to the total atom number $\iint \text{d}x\text{d}y\, |\psi(x, y, t)|^2 = N$.

To solve Eq. (4), we employ a split-step Fourier method with grid resolution $N_x = N_y = 256$, a spatial size $L_x = L_y = 100l_r$, and a time step $\Delta t = 2.5 \times 10^{-6}\omega_r^{-1}$, where $l_r = \sqrt{\hbar/M\omega_r}$ is the radial harmonic oscillator length. We have verified that our results are unchanged after doubling the number of grid points in each direction.

### 2.2 Accelerating double-ring system with a closed barrier

To understand the role of acceleration on our system, we first investigate the effect of acceleration in a double-ring system while keeping the barrier closed ($V_0 = 0$) with an antivortex imprinted into the leading ring, with respect to the acceleration direction. A numerical

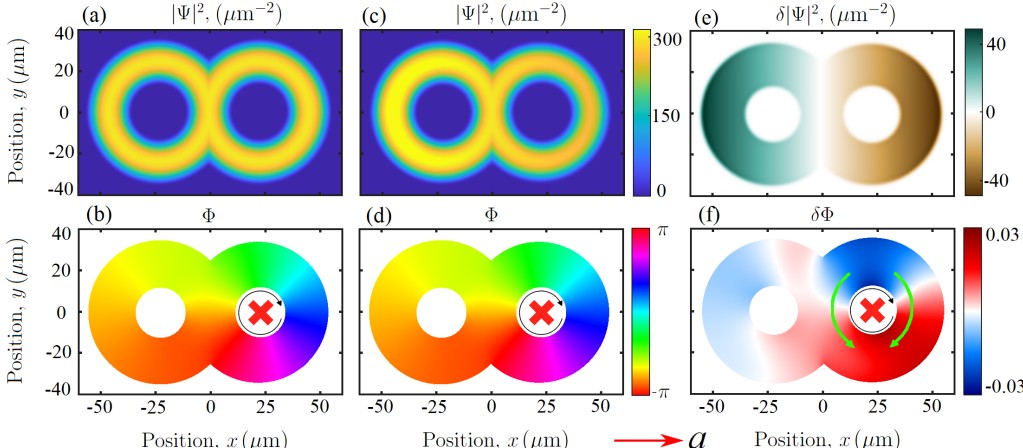

Figure 1: Effect of acceleration on a double-ring system with a persistent current. (a) Density $n = |\psi|^2$ and (b) phase $\Phi$ distributions at zero acceleration. (c) and (d) show the density and phase under horizontal acceleration $a = 0.01g$ in the positive $x$-direction. To visualize the induced changes, (e) and (f) present the relative differences between the columns. The red cross marks the antivortex position, and black arrows indicate the flow direction induced by the vortex. In (f), green arrows show the additional flow induced by acceleration in the leading ring.

demonstration of the role of acceleration is shown in the second column of Fig. 1, for a specific example of $a = a_x = 0.01\,g$ (relative to Earth's gravity, $g = 9.81$ m/s$^2$), taken at $t = 0$, after solving Eq. (4) in imaginary time. This highlights that a key modification compared to the previously studied static case, shown in the first column, is a density redistribution. The general behavior is quite clear from the Thomas-Fermi solution to Eq. (4), where the density is lower in the forwardmost part of the system. This is reminiscent of the classical situation, in how a water surface inclines in a water tank moving with constant acceleration.

In the presence of a persistent current, the density variation also leads to a phase variation away from the usual azimuthal $2\pi$ winding, as shown in Fig. 1(f), where the phase is plotted relative to the static case. This additional phase variation appears in order to satisfy the hydrodynamic continuity equation. Hence, the flow velocity is greater in a less dense region–the forwardmost part of the system–and here we have faster phase accumulation. This build-up of phase at the bottom of the leading ring resembles a Magnus-like effect [35], and if we instead imprint a vortex with positive circulation, this effect flips about the $y$-axis. This phase effect also occurs for a single ring under acceleration.

As the acceleration depletes the front part of the leading ring, a sufficiently large acceleration will lead to a near-zero density in the forwardmost part, thus disrupting our studied two-ring topology. This, in turn, sets an upper bound to the acceleration, as the vortex simply leaves the system. For our example parameters, this critical acceleration is around $a_{\mathrm{cr}} \sim 0.06\,g$. Remarkably, the results we present in this work are independent of the instantaneous velocity, provided that it is lower than the speed of sound.

## 2.3 Asymmetric undamped persistent current oscillations

In the recent work of Ref. [32], it was shown that in the static double-ring system with a single persistent current, an introduction of a weak link with amplitude exceeding the system chemical potential ($V_0 > \mu_{2D}$) will initiate an oscillation of this current between the rings, that persists indefinitely at zero temperature. After allowing the closed system to evolve for an initial 100 ms, the potential barrier is linearly ramped up to a maximum value of $V_0 = 1.2\mu_{2D}$

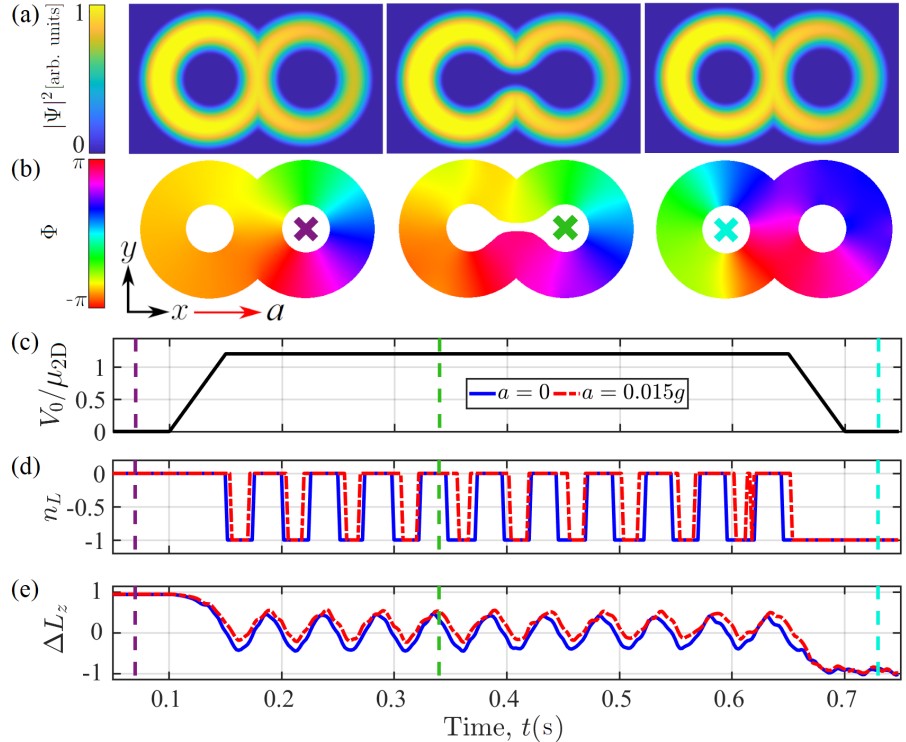

Figure 2: Persistent current oscillations in accelerating rings. The density (a) and phase (b) evolution of the double ring under constant acceleration $a = 0.015g$, directed along the $x$-axis. The phase shows the vortex transition from the right to the left ring. (c) Barrier amplitude as a function of time. (d) Time evolution of winding number for the left ring $n_L$. (e) Angular momentum difference dynamics between rings. The dashed red line represents data for acceleration $a = 0.015g$, compared to the solid blue line of zero acceleration. Vertical dashed lines in (c)-(e) highlight the times at which the corresponding density (a) and phase (b) profiles were plotted, colored to match the corresponding antivortex cross.

[Fig. 2(c)]. When the barrier's amplitude exceeds the chemical potential, the system topology transforms to the torus, allowing vortex transitions, and so oscillations of the persistent current between the two rings.

Here, we study the influence of acceleration on such persistent current oscillations under the same protocol. An exemplary case is shown in Fig. 2, for fixed horizontal acceleration $a = a_x = 0.015g$. To measure the vortex position with time, we used two quantities: the winding phase around the ring with radius $R$, as measured from the center of each ring, and the angular momentum per particle difference in each ring. The latter quantity is defined as $\Delta L_z = \langle L_{z,L} \rangle - \langle L_{z,R} \rangle$, where

$$\langle L_{z,\{L,R\}} \rangle = \frac{i\hbar}{N_{\{L,R\}}} \iint_{\mathcal{R}} \mathrm{d}x \mathrm{d}y\, \psi^* \left( y \frac{\partial}{\partial x} - (x \pm R) \frac{\partial}{\partial y} \right) \psi \,. \tag{5}$$

Here, $N_{L/R}$ is the particle number in the left/right ring respectively, and $\mathcal{R}$ denotes the corresponding integration region of the left/right part of the dimer.

For small accelerations $a \ll g$ (not shown), the effect of acceleration on the oscillations is subtle, and the overall picture is quite similar to the case without acceleration [32]. With a larger acceleration, the vortex spends more time in the leading ring, while the total period weakly changes. In the measured winding number in the left ring $n_L$ [Fig. 2(d)] this presents

as an asymmetry in the time spent by the vortex in the left versus right ring. For the angular momentum difference [Fig. 2(e)] this creates an offset in the curve such that $\Delta L_z$ oscillates around a non-zero constant.

Such a bias can be explained by considering the ground state for the open gate part of the protocol. Similar to how density and phase redistribute for the closed system [Fig. 1], for the open ring the density in the leading (right) part is lower, and phase accumulation there is faster. However, in this case, the overall $2\pi$ phase is distributed unequally around the two rings; the phase circulation around the leading ring is slightly higher due to acceleration. Therefore, one can say that a vortex is slightly more "localized" in the leading ring. In the full dynamic protocol, the system's energy is higher than the ground state, and consequently the vortex oscillates between rings. This argumentation, however, naturally explains the observed preference for the vortex to be found in the leading ring.

The same result can also be obtained from the angular momentum per particle difference $\Delta L_z$. Considering the overall system in the form of a "thin wire", with constant negligible width $\sigma$, we can estimate the angular momentum difference in the following way:

$$\Delta L = \frac{\tilde{L}_L}{N_L} - \frac{\tilde{L}_R}{N_R}, \tag{6}$$

where $\tilde{L}_i = j\sigma \int_{l_i} dl\, r(l)$ is the total angular momentum, around the given ring. Here, $j$ is the current density, which is constant along the wire, and the integral denotes integration over the full contour of the distance $r(l)$ from the center of the ring to a wire. Importantly, we get the same value for both contours if they are symmetric, so $\tilde{L}_L = \tilde{L}_R$, and this value depends only on geometry. Thus, for small acceleration, we obtain $\Delta L \sim N_R - N_L \sim -q a_x$, utilizing the Thomas-Fermi solution that shows the effect of acceleration on the density is a redistribution of atoms linearly proportional to $a_x$, and $q = \pm 1$ is the vortex/antivortex charge.[1] Hence, we can say that each ring has the same total angular momentum, but the angular momentum per particle is bigger in the forward ring, due to the lower number of particles there.

Our simplifying analytical argumentation shows that at small acceleration, the angular momentum per particle difference $\Delta L$ depends mainly linearly on $a_x$, verifying the numerical simulations. Crucially, this acceleration dependence provides an observable that we will later exploit when discussing designs for an accelerometer.

In Fig. 2, the time at which the barrier ramp-down sequence is initiated affects the final position of the vortex. In the static case, it is known that the oscillations halt as soon as $V_0$ becomes smaller than $\mu_{2D}$. As we will discuss later in the damped regime [Fig. 4], the critical barrier amplitude required to suppress oscillations may be even lower under acceleration. If the goal is to control the final state of the vortex, then this can be achieved by calibrating the maximum barrier amplitude to be as close to the chemical potential as possible. Furthermore, since the oscillation period is independent of acceleration, the optimal barrier closing time can be predetermined.

## 3  Effect of (thermal-induced) dissipation

As discussed in detail in our previous work [32], dissipation due to thermal fluctuations [36–39] can affect the persistent current oscillations to such extent, that it can even suppress oscillations between rings. Our earlier work has shown that the dominant features in a dissipative system are qualitatively captured by a simple phenomenological model [40, 41], widely used

---

[1]Note that for a vortex ($q = 1$), $\Delta L < 0$ means localization in the right ring, whereas for an antivortex ($q = -1$) this is instead given by $\Delta L > 0$.

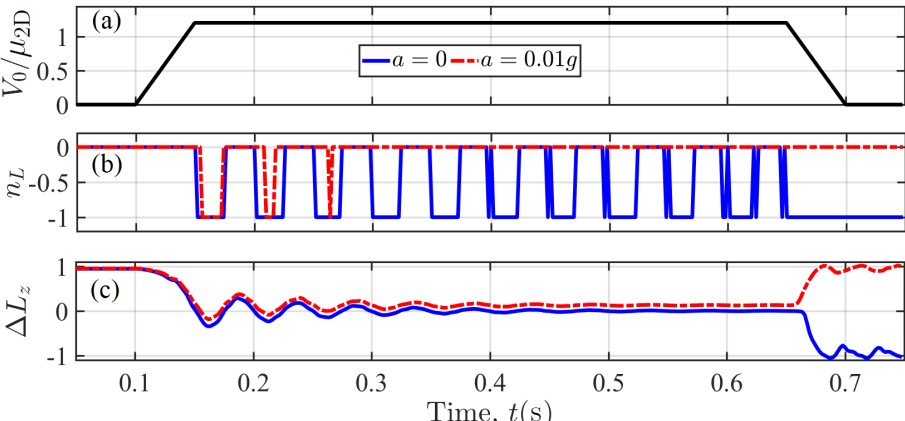

Figure 3: Oscillation decay under small dissipation $\gamma = 0.001$. The barrier protocol (a) is the same as in Fig. 2. (b) Winding number in the left ring for zero acceleration (blue solid curve) and $a = 0.01g$ (red dashed curve). (c) Angular momentum per particle difference. Note that in the case of zero acceleration, the final placement of the vortex does not prefer any ring, depending only on protocol time parameters.

in the literature due to its inherent simplicity, in which the dominant role of thermal dissipation is modelled through the addition of a dissipative factor $0 < \gamma \ll 1$ in the Gross-Pitaevskii equation. This term signifies the coupling between the condensate atoms and thermal cloud, resulting in a dissipative GPE in the presence of external acceleration that takes the form[2]

$$(i - \gamma)\hbar \frac{\partial \psi}{\partial t} = \left( -\frac{\hbar^2}{2M}\nabla^2 + V_{\text{ext}} + g_{2D}|\psi|^2 + M\boldsymbol{a} \cdot \mathbf{r} - \tilde{\mu}_{2D} \right)\psi. \tag{7}$$

A single real time simulation with finite $\gamma$ can, in some sense, be considered equivalent to the ensemble average of many experimental runs, i.e. averaging out thermal fluctuations but keeping the overall energy reducing effect [37, 45–47], an effect explicitly demonstrated for dark soliton dynamics in Refs. [48, 49]. In such a way, all excitations decay in time, and the wave function tends to a stationary state. With the inclusion of thermal fluctuations, such a setup has been used to model observations pertaining to persistent current formation and dynamics [50–52]. We note that, for our simulations, the chemical potential is adjusted at each time step providing total atom number conservation, as was employed in Refs. [41, 53, 54].

Note that the introduction of a phenomenological dissipation in this manner implicitly models the dissipation within the context of an accelerating frame: the thermal cloud is implicitly co-moving with the condensate. In principle, one could alternatively implement the dissipation in the laboratory frame, by explicitly moving the trap with constant acceleration. However, this would result in the dissipation being implicitly modelled within the context of the laboratory frame, and would lead to a completely different qualitative behavior of the vortex transitions for our system. Such a scenario is not considered further in the main text, as we assume the thermal cloud moves with the potential. A detailed discussion of this alternate case is provided in the Appendix.

## 3.1 Weakly dissipative regime

In the case of zero acceleration and weak dissipation ($\gamma < \gamma_c$), the $\Delta L_z$ oscillation amplitude decays over time, eventually leading to the vortex becoming trapped at the system's

---

[2]In some other works [42–44] authors rather replace the Hamiltonian $\hat{H} \to (1 - i\Lambda)\hat{H}$, however, for small $\gamma$ and $\Lambda$ these approaches are the same.

center [32]. The oscillation's lifetime decreases with higher $\gamma$, and, at some critical value $\gamma_c = 0.015$, the transport of persistent current seizes, i.e. the vortex stays in the initial ring.

As was shown in Fig. 2, the acceleration creates a bias in the angular momentum distribution between rings. This feature becomes crucial under dissipation, as explored in Fig. 3, as this forces the equilibrium final position of the vortex within the leading ring. The oscillation amplitude $\Delta L$ dissipates not to zero, as for zero acceleration, but to some constant value [Fig. 3(c)] in accordance with Eq. (6). When the oscillation amplitude of $\Delta L$ decays to a smaller value than this constant offset, the vortex partially localizes to the leading ring, and closing the barrier finishes this localization. Note that this is independent of the choice of the ring where the persistent current is imprinted or the sign of the vortex charge; however, the vortex will always reside in the leading ring, providing a measurement of the sign of $a_x$. We find that larger acceleration halts oscillations earlier, for fixed $\gamma$, as less time is needed for the initial amplitude of angular momentum difference to decay below the bias amplitude.

## 3.2 Strongly dissipative regime

In this section, we examine the specific case of a high dissipation rate ($\gamma > \gamma_c$), where persistent current oscillations do not occur. This regime is distinctive due to the presence of unambiguous transitions of the persistent currents between the two rings: any observed transition must be due to external forces.

As previously mentioned, under large damping the vortex relaxes at either the system center if static, or in the leading ring if accelerating, provided that the barrier's amplitude exceeds the chemical potential. We can prevent this, however, by lowering the barrier amplitude. Indeed, for fixed $V_0 \lesssim \mu_{2D}$ a transition can be seeded by overcoming the remaining energetic barrier through acceleration. In such a way, we can control vortex transitions in the system, obtaining a detailed understanding of where the vortex should ultimately reside for a given acceleration and barrier's amplitude. This key prediction paves the way for the acceleration measurement.

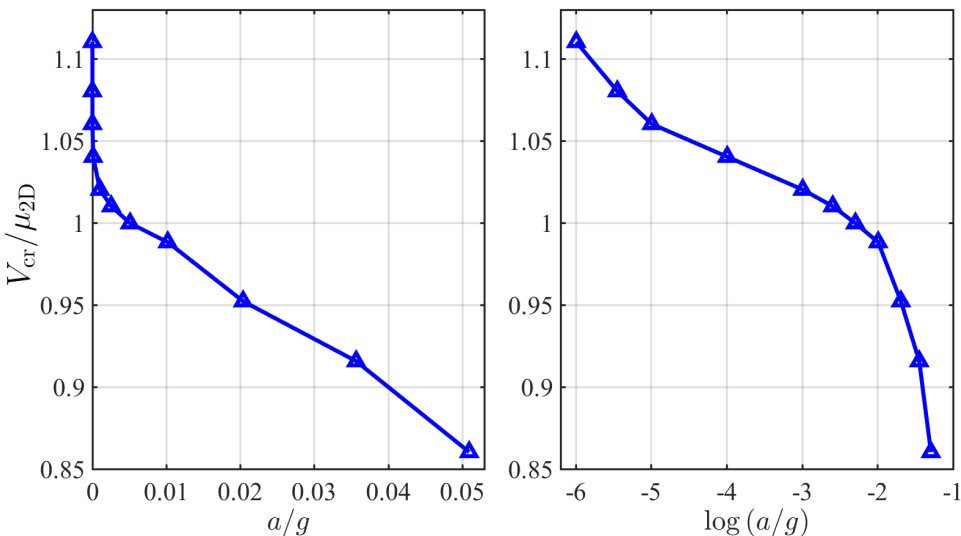

Figure 4: Critical barrier amplitude for vortex transfer in the strongly damped regime $\gamma = 0.02$, for a given longitudinal acceleration and states with an initial single vortex imbalance. The left side shows data on a linear scale, and the right shows the same data on a semilogarithmic scale. The precision on the $y$-axis is around 10 Hz ($0.005\mu_{2D}$).

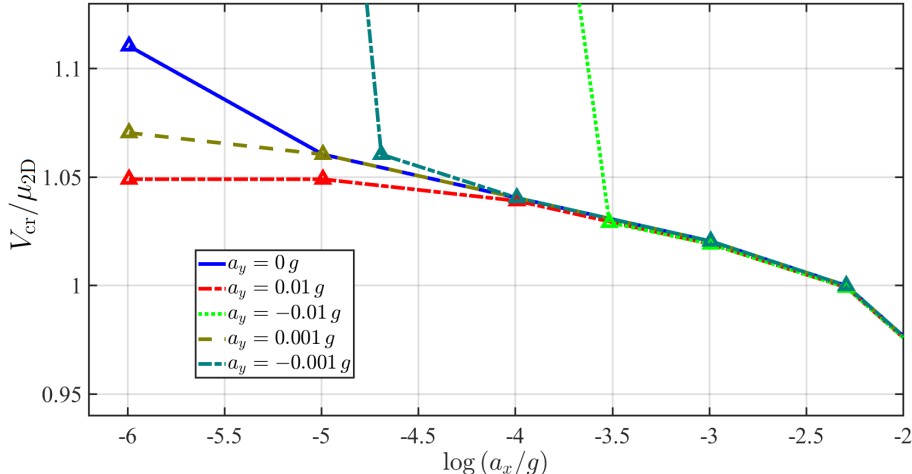

Figure 5: Critical barrier dependency by longitudinal acceleration ($a_x$) to transfer from (-1,0) to (0,-1), with a fixed transverse acceleration component ($a_y$) in the semi-log scale and $\gamma = 0.02$. The precision on the $y$-axis is around 10 Hz.

Figure 4 shows the critical energetic barrier amplitude $V_{\mathrm{cr}}$ required to facilitate vortex transfer from the trailing to the leading ring for a given acceleration, revealing a monotonically varying function in terms of acceleration. As there is a unique critical value for each acceleration, one can devise a protocol to measure the acceleration. Under conditions of continuous acceleration, linearly increasing the amplitude of the potential barrier and measuring the moment the vortex transfers, through continuous minimally-destructive measurement of the vortex position, will return an estimate of the acceleration. Such non-intrusive protocols for monitoring the persistent current include measuring the Doppler shift of phonons [55] or measuring the directionality of atoms leaking into an external channel [56].

## 4 Acceleration in the plane

As we have shown, the longitudinal acceleration component ($a_x$) distinguishes the two rings, causing a density bias in the trailing ring and a transverse phase shift perpendicular to the acceleration. While one might expect the transverse acceleration to have no effect on the vortex transition, as it equally modifies both rings' density, the phase deviation in this case occurs in the longitudinal plane. This can either drive or inhibit vortex transitions, depending on the vortex sign or direction of travel.

To investigate this, we numerically study how adding a transverse component to the acceleration affects the critical barrier amplitude for transition in the over-damped scenario. Our results reveal two regimes, as is evident in Fig. 5. If $a_x \geq 10^{-3.5} g$, then the choice of $a_y$ does not significantly change the critical barrier amplitude, as long as acceleration does not disrupt the two-ring geometry ($a \lesssim a_{\mathrm{cr}}$). However, for smaller $a_x$, it seems that the sign and magnitude of $a_y$ can significantly impact the critical barrier amplitude. If $a_y \gg a_x > 0$, the transverse acceleration reduces the critical barrier amplitude, and, most intriguingly, if $a_y \ll -a_x < 0$ the transition can be completely suppressed. These results can be readily understood from the perpendicular phase effect discussed in Fig. 1 (f): when the angle of attack is transverse to the ring's orientation (when $\left| \arctan(a_x/a_y) \right| < 1°$) this bias suppresses the transition of an antivortex. The nature of this effect is dependent on the sign of the vortex circulation. Given the directional dependence of the transverse Magnus force with the sign of the circulation, it can be demonstrated that the transition for a vortex with given $a_x$ and $a_y$ occurs in the same

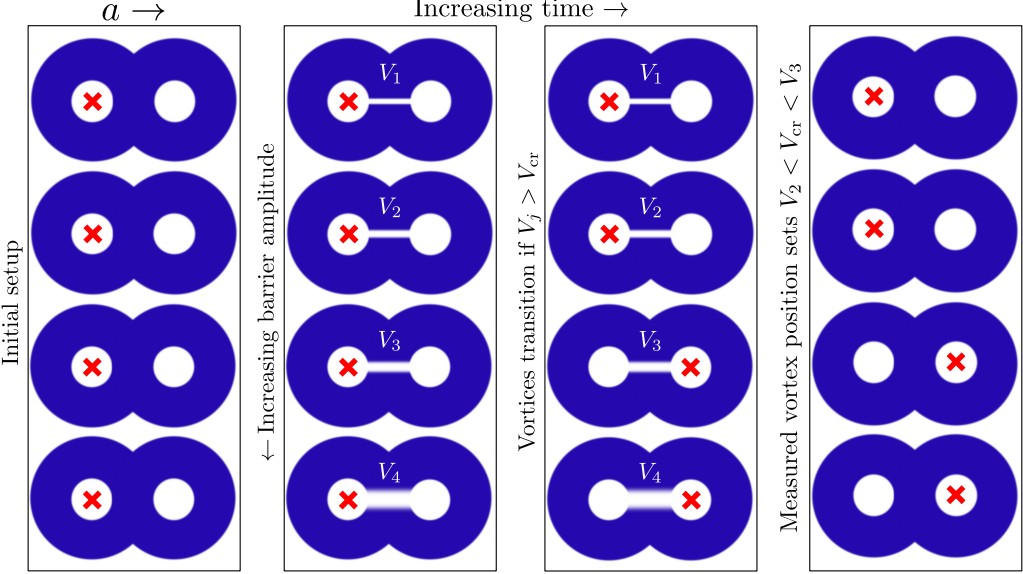

Figure 6: Schematic of a quantum gas accelerometer. Initially, each sub-system of double-rings is initialized with a single vortex. From top-to-bottom, the gates are opened with increasing amplitude. After some time of linear rightward acceleration, the barriers are removed and the final vortex position is measured. Where the transition has occurred sets the range of possible values of the critical barrier amplitude, and therefore the value of the acceleration.

manner for an antivortex with the same longitudinal acceleration $a_x$, but with opposite transverse acceleration $-a_y$. This tool gives another layer of control when employing the results here to an accelerometer.

Finally, we note that the aforementioned critical barrier and critical angles are slightly sensitive to the value of the present dissipation $\gamma$, with an especially pronounced difference for $\gamma > 0.03$. However, such values of $\gamma$ are likely not very physically relevant in ultracold atomic gases [46, 57–60]. The impact of the longitudinal component in other dissipative regimes of smaller $\gamma$ remains unchanged.

# 5 A proof-of-concept accelerometer

We have shown thus far that the transition of a vortex from the trailing to leading ring in the direction of acceleration is a sensitive probe to the acceleration vector, and if the barrier amplitude at the time of transfer is known the acceleration can be read out. In a single double-ring system, this can be achieved by continuous non-invasive measurement of the persistent current [55, 56] whilst ramping up the barrier amplitude.

We propose that in order to use these results to measure the acceleration, a simpler protocol requires initializing many double-ring systems simultaneously, and performing a single destructive measurement of the vortex position. A schematic setup to measure linear acceleration in a single experimental run is demonstrated in Fig. 6. By using multiple independent double-rings, a single measurement can place a bound on the critical barrier amplitude. The limitation of the accuracy of the measurement is entirely set by the tightness of this bound, and repeat measurements–narrowing the range of maximum barrier amplitudes probed–can further improve this. To facilitate this destructive measurement, each ring can have a central reference condensate to create a winding-dependent spiral interference pattern through time-

of-flight expansion [17,19,61,62]. Alternatively, a straightforward time-of-flight measurement of the radial size of the hole created by the persistent current can be performed [63].

By varying the angle of neighboring double-ring systems (such that they are positioned in a circle, like the hour marks on a clock, for example) and implementing systems with both positive and negatively charged vortices, one should be able to probe the full planar acceleration on a wide range of scales, while also avoiding the 'blind-spot' when the acceleration is perpendicular to the direction of travel.

Ultracold atom-based accelerometers and gravimeters have demonstrated sensitivity to accelerations in the range of $10^{-9}g$ to $10^{-5}g$ [25–30]. Our results indicate that we can resolve accelerations down to $10^{-6}g$. With the same methods, even higher sensitivity is achievable, but resolving the critical barrier amplitude at such low accelerations requires a finer spatiotemporal grid than used here. Furthermore, down at this level, fluctuations will start to play an important role. In future work, we will test and characterize the quality of our atomtronic accelerometer, to provide limits on the potential sensitivity of this new quantum technology.

## 6 Conclusion

In this work, we have studied the influence of acceleration on the persistent current transfer across two rings with a tunable weak link in trapped quasi-2D quantum gases. Whilst the barrier is open, our 2D simulations and qualitative arguments showed that acceleration drives a bias in the vortex transitions. In the absence of dissipation, the vortex oscillates between rings but prefers to stay longer in the leading ring, with respect to the direction of acceleration. However, such a difference can only be discerned for a higher acceleration rate. Under dissipation, the oscillations decay, and the vortex settles within the leading ring.

The final control parameter to exploit for acceleration measurements is the controllable weak-link barrier amplitude. When the damping is strong enough to suppress vortex oscillations, vortex transmission occurs only if the acceleration exceeds a certain threshold, determined by a critical amplitude of the barrier potential. Thus, obtaining this critical value unambiguously characterizes linear acceleration and provides a promising platform for acceleration measurement. Furthermore, our numerical studies and qualitative analyses show that the system is mainly sensitive to the acceleration projection on the main axis of the dimer, so one only needs to measure each acceleration projection independently to obtain all the information about the direction and amplitude of the linear acceleration. Using these results, we have designed a first proof-of-principle accelerometer that can operate in the range $-6 < \log(a/g) < -1$, however we believe that the lower bound can be further reduced. In future studies, we will place rigorous bounds on the efficacy of double-ring quantum gas accelerometers.

## Acknowledgments

We acknowledge Oksana Chelpanova for useful discussions. N.P, M.E., A.Y and T.B. acknowledge the Benasque atomtronics workshop series held at Centro de Ciencias de Benasque Pedro Pascual, which have provided the opportunities for initiating our collaborative research.

**Funding information** A.C. and A.O. acknowledge support from the National Research Foundation of Ukraine through Grant No. 2020.02/0032. A.Y. acknowledges support from the projects 'Ultracold atoms in curved geometries', 'Theoretical analysis of quantum atomic mixtures' of the University of Padova, and from INFN. M.E. acknowledges support from US Na-

tional Science Foundation grant number PHY-2207476. T.B. acknowledges financial support by the ESQ Discovery programme (Erwin Schrödinger Center for Quantum Science & Technology), hosted by the Austrian Academy of Sciences (ÖAW), the European Research Council through the Advanced Grant DyMETEr (10.3030/101054500), and funding from FWF Grant No. I4426. N.P. acknowledges the Quantera ERA-NET cofund project NAQUAS through the Engineering and Physical Science Research Council, Grant No. EP/R043434/1 during early stages of this project.

# A  Damped GPE in the non-inertial frame

To get a governing equation in the non-inertial frame, one can use the approach detailed in Ref. [64]. We consider the transformation to an accelerated frame, starting from the dissipative GPE introduced in the laboratory frame with an explicitly moving trap

$$(i-\gamma)\hbar\frac{\partial\Psi}{\partial t}=\left(-\frac{h^2}{2M}\nabla^2+V(\boldsymbol{x},t)+g|\Psi|^2-\mu\right)\Psi. \tag{A.1}$$

We use the following coordinate transformation to obtain an accelerated frame

$$\boldsymbol{x}'=\boldsymbol{x}-\int_0^t\boldsymbol{v}(t')\,dt'=\boldsymbol{x}-\tilde{\boldsymbol{x}}(t),$$

where the moving potential has the stationary form

$$V(\boldsymbol{x},t)=V(\boldsymbol{x}').$$

Firstly, we want to rewrite Eq. (A.1) for the wave function in the accelerated frame $\Psi'(\boldsymbol{x}',\tau)$. So $\Psi'(\boldsymbol{x}',\tau)=\Psi(\boldsymbol{x},t)=\Psi(\boldsymbol{x}'+\tilde{\boldsymbol{x}}(\tau),\tau)$. We note,

$$\frac{\partial}{\partial t}\Psi(\boldsymbol{x},t)=\frac{\partial}{\partial t}\Psi'(\boldsymbol{x}',\tau)=\left[\frac{\partial}{\partial\tau}-\tilde{\boldsymbol{v}}(\tau)\cdot\boldsymbol{\nabla}'\right]\Psi'(\boldsymbol{x}',\tau),$$
$$\nabla\Psi(\boldsymbol{x},t)=\nabla'\Psi'(\boldsymbol{x}',\tau).$$

Using these transformations for Eq. (A.1) one gets the following equation for the wave function in the moving frame

$$(i-\gamma)\hbar\frac{\partial\Psi'(\boldsymbol{x}',\tau)}{\partial\tau}$$
$$=\left[\frac{\left(-i\hbar\nabla'-M\tilde{\boldsymbol{v}}(\tau)\right)^2}{2M}+V(\boldsymbol{x}')+g|\Psi'(\boldsymbol{x}',\tau)|^2-\frac{M\tilde{v}^2(\tau)}{2}-\mu-\gamma\hbar\tilde{\boldsymbol{v}}(\tau)\cdot\boldsymbol{\nabla}'\right]\Psi'(\boldsymbol{x}',\tau).$$

Secondly, we need to apply the following gauge transformation

$$\Psi'(\boldsymbol{x}',\tau)=e^{\frac{iM}{\hbar}\left[\tilde{\boldsymbol{v}}(\tau)\cdot\boldsymbol{x}'+\frac{1}{2}\int_0^\tau v(t)^2\,dt\right]}\psi(\boldsymbol{x}',\tau),$$

where $\tilde{\boldsymbol{v}}$ and $\tilde{\boldsymbol{a}}$, the velocity and acceleration of the moving potential, respectively, are time derivatives of $\tilde{\boldsymbol{x}}$.

After straightforward algebra, one gets the following equation for $\psi(\boldsymbol{x}',\tau)$

$$(i-\gamma)\hbar\frac{\partial\psi(\boldsymbol{x}',\tau)}{\partial\tau}=\left[-\frac{\hbar^2}{2M}\nabla'^2+V(\boldsymbol{x}')+M\tilde{\boldsymbol{a}}(\tau)\cdot\boldsymbol{x}'+g|\psi|^2-\mu\right]\psi(\boldsymbol{x}',\tau)$$
$$+i\gamma\left[M\tilde{\boldsymbol{a}}(\tau)\cdot\boldsymbol{x}'-M\frac{\tilde{v}^2(\tau)}{2}+i\hbar\tilde{\boldsymbol{v}}(\tau)\cdot\boldsymbol{\nabla}'\right]\psi(\boldsymbol{x}',\tau). \tag{A.2}$$

Taking $\gamma = 0$, we restore the conservative GPE in the accelerated frame, Eq. (4).

We see that such an approach gives a different equation for the wave function when compared to Eq. (7), where the dissipation was introduced only after moving to the accelerated frame. Here, we have additional explicitly time-dependent terms, for non-zero acceleration, proportional to the effective dissipation $\gamma$. Considering the influence of these terms, from a hydrodynamic perspective, one can interpret them as some kind of friction between condensate and thermal atoms, which redistributes and induces flow in the system, similar to wind. Now that the right-hand side is time-dependent, there is no final stationary state to which the system relaxes. Such a scenario is not particularly realistic, as a thermal cloud typically remains trapped together with the condensate, and so moves with it.

We note that the inclusion of dissipation arises in various types of trap motion. For rotating traps, the literature presents two approaches: dissipation can be included in the rotating frame [41,65], or in the laboratory frame, by explicitly incorporating a rotating potential [62,66,67]. The rotating frame approach is more commonly used. As demonstrated in Ref. [68], these approaches are not equivalent—the dissipation terms are tied to the frame in which they are implemented. For constant rotation, both approaches yield a time-independent Hamiltonian in the rotating frame, but only dissipation terms implemented in the rotating frame allow relaxation to the same ground state as in the conservative case. Therefore, only in the moving frame can a simple phenomenological dissipation model accurately describe proper relaxation.

## B  Multiply-charged vortex transitions

The focus of our work has been on the effect of acceleration on the oscillation of a single persistent current. Nonetheless, we can also generalize our results to the dynamics in the context of multiple windings. Hereafter, we use the following shorthand of the system's state as $(m, n)$, where $m$ $(n)$ denotes the integer number of vortices in the left (right) ring, i.e. the trailing (leading) ring.

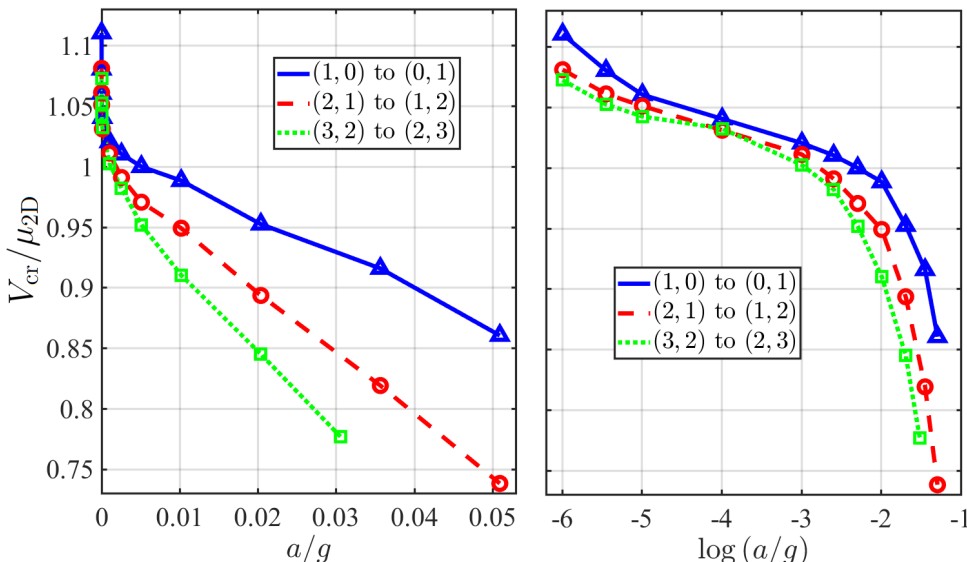

Figure 7: Critical barrier amplitude for vortex transfer in the strongly damped regime $\gamma = 0.02$, for a given longitudinal acceleration and states with an initial single vortex imbalance. The left side shows data on a linear scale, and the right shows the same data on a semilogarithmic scale. The precision on the $y$-axis is around 10 Hz.

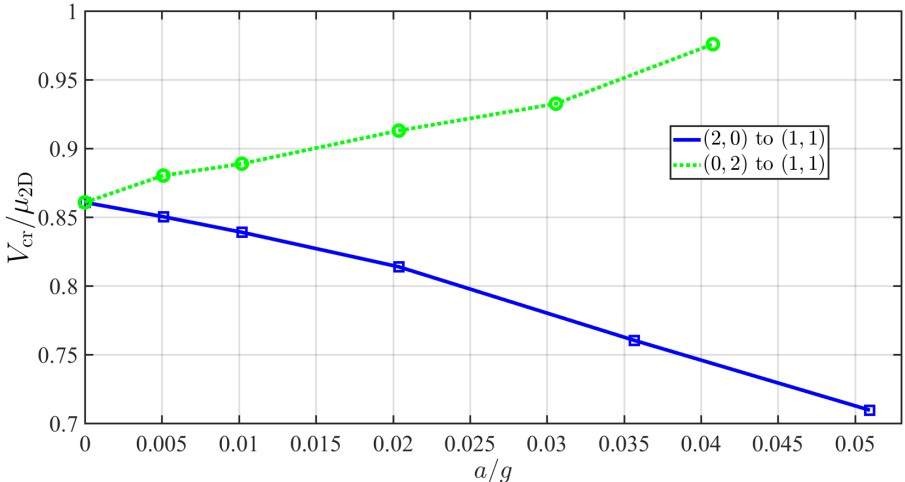

Figure 8: Critical barrier dependence on longitudinal acceleration for two vortices within a system. Parameters as in the main text, with $\gamma = 0.02$. The precision on the $y$-axis is around 10 Hz.

Oscillations, or single transitions in the strongly damped regime, where the initial imbalance between rings is a single vortex, $(m + 1, m)$, should behave similarly regardless of $m$. Without acceleration, there is no energy preference between the initial and final states. However, due to the higher kinetic energy of states with larger $m$, transitions at fixed $V_{cr}$ should occur at smaller accelerations. This hypothesis is supported by our simulations, as shown in Fig. 7.[3] In general, increasing the total winding number decreases the critical barrier amplitude, an effect that is significantly more pronounced at larger accelerations.

Combinations where the initial vortex imbalance is two provides a unique probe into the critical barrier frequency. In this case, transitions can happen in the absence of acceleration, even when the amplitude is well below the chemical potential. The stationary state here is to spread the initial imbalance from either $(m, m + 2)$ or $(m + 2, m)$ to $(m + 1, m + 1)$. However, the acceleration inclusion deforms the necessary critical amplitude for the transition in a way that is dependent on the direction of acceleration, or, equivalently, which ring is initiated with the larger persistent current.

We explore this effect comparing the initial states (2,0) and (0,2) in Fig. 8. Given the general tendency for the vortex to localize in the leading ring, the dependency on acceleration becomes clear. For the initial state (2,0), increasing the acceleration reduces the energy cost for the vortex to transit. However, if the initial state is (0,2), both vortices are already in the leading ring, and instead a larger barrier amplitude is required to facilitate the transfer to (1,1). Moreover, for acceleration $a > 0.04g$ the bias is so high, that there are no transitions at all. In this case, a detailed analysis of the angular momentum difference shows that, when the barrier is open, more than 75% of the angular momentum accumulated in the front ring, hence more than "1.5" vortices are already in the front part of a system, and $(0, 2)$ is considered as a new equilibrium state. We note that these observations are general for larger total angular momentum, but with a global shift to lower critical barrier frequency, similar to Fig. 7.

The behavior of other transitions for many-vortex states can be inferred from these examples: if $m + n$ is even the system will distribute the angular momentum equally, otherwise if it is odd there will be an additional current that transfers to the leading ring. Finally, if the winding in each ring initially has opposite charge, the vortices first annihilate $[(−m, m ± n) \rightarrow (0, ±n)]$ before the remainder behave with the rules described above.

---

[3]Note the data for the (1,0) to (0,1) transition is the same as Fig. 4.

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
