# Peer review of "Acceleration-induced transport of quantum vortices in joined atomtronic circuits"

_SciPost Physics, doi:SciPost Phys. 19, 005 (2025)_

## Round 1 · Referee Report · Anonymous (Referee 1) · 2025-4-22

Strengths
One advantage of using guided ultracold atoms for quantum sensing, as opposed to free-space interferometers, is that the setup can be made more compact and portable. This is a great advantage which makes this proposal competitive, despite its predicted sensitivity not reaching the best sensitivities of state-of-the-art accelerometers.
Weaknesses
Report
The paper is highly original and the results are presented very clearly. I only have a few requested changes, detailed below.
Requested changes
1) The introduction is exhaustive, however I think Dowling's proposal for rotation sensing with a superposition of persistent currents should be included. See: https://arxiv.org/abs/0907.1138
2) In Fig 1, for panels (a) and (b) at t=0, is it worth also stating explicitly that a=0?
2) In Fig 2, where persistent current oscillations are shown, is there a best time within the oscillation for V_0 to begin the ramp to 0, so that the vortex is transferred? If the ramp starts at the wrong time, will the vortex go back to the leading ring?
Recommendation
Publish (surpasses expectations and criteria for this Journal; among top 10%)

Author: Thomas Bland on 2025-05-20 [id 5498]
(in reply to Report 1 on 2025-04-22)We thank the Referee for their report, and are delighted that they are excited by this work as much as we are. Below, we respond to the three listed requested changes by the Referee:
1) The introduction is exhaustive, however I think Dowling's proposal for rotation sensing with a superposition of persistent currents should be included. See: https://arxiv.org/abs/0907.1138
We thank the Referee for highlighting this relevant work. We will include this reference in the next version of the manuscript.
2) In Fig 1, for panels (a) and (b) at t=0, is it worth also stating explicitly that a=0?
The data shown in Fig. 1 were obtained with a finite moving frame acceleration of $a = 0.01g$. Panels (a) and (b) were generated using the imaginary time technique, which yields the stationary solution for a fixed acceleration. These are then compared to the static case ($a = 0$), whose solutions--while not displayed--were discussed in detail in our previous work Phys. Rev. Res. 4, 043171 (2022). Panels (c) and (d) show the differences between the static and accelerated cases.
To improve clarity and eliminate the need for referencing earlier work, we will revise Fig. 1 to include three columns: the non-accelerating case, the accelerating case, and their differences. We believe this presentation will be more self-contained and accessible for the reader.
3) In Fig 2, where persistent current oscillations are shown, is there a best time within the oscillation for $V_0$ to begin the ramp to 0, so that the vortex is transferred? If the ramp starts at the wrong time, will the vortex go back to the leading ring?
We thank the Referee for this insightful question. The answer is two-fold. First, the ramp rate has minimal impact on the dynamics, except in the case of an instantaneous ramp. This is because vortex oscillations are suppressed as soon as $V < \mu$, which typically happens shortly after the ramp begins, effectively preventing the scenario raised by the Referee. In the absence of acceleration, this behavior was discussed in our previous work Phys. Rev. Res. 4, 043171 (2022).
However, under acceleration, this warrants further analysis: the critical barrier amplitude becomes acceleration-dependent, altering the timing between initiating the barrier closure and suppressing vortex oscillations. This can be addressed by calibrating the system such that the initial barrier height is as close to $V/\mu = 1$ as possible. Moreover, since the oscillation period is independent of acceleration, it is possible to pre-determine an optimal time to begin the ramp. We will incorporate this discussion into the revised manuscript.
Best wishes,
Thomas Bland on behalf of all authors

---

## Round 1 · Referee Report · Anonymous (Referee 2) · 2025-6-3

Strengths
Weaknesses
Report
In Section 2, the Authors consider the dissipationless case of a double ring. The modeling is done based on the 2D Gross-Pitaevskii Equation. The axis of the double ring is the x axis and they consider acceleration along that direction, ax.
In a previous work, [32], they found that in a static double ring system with a weak link with a large enough amplitude, the current oscillates between the rings. Now, they carry out calculations for the case of a time-dependent barrier height. When the barrier height is larger, the vortex, basically, goes back and forth between the rings.
In Section 3, they consider the case of dissipation. They incorporate dissipation with added gamma constant in the dynamical equation following the literature. For gamma<gamma_c, the vortex oscillation takes place, which they study as the weakly dissipative regime. The oscillation lifetime decreases with an increasing gamma. The oscillation becomes less regular. For gamma>gamma_c, the vortex oscillation stops, which they study as the strongly dissipative regime.
In Section 4, the authors consider acceleration a_y, which is not in the direction of the axis of the double ring, but in an orthogonal direction in plane. They find that the value of ay influences the response of the setup to ax, especially, if ax is small. It has less influence if ax is larger.
In Section 5 and in Figure 6, a proof-of-concept accelerometer is outlined. An array of double-rings are used and the acceleration takes places parallel to the axis of the double rings. In the protocol, the double-rings are initialized with a single vortex. Then, the gates in the double rings are opened with an increasing amplitude. After some time, the barriers are removed and the position of the vortex is detected. In some double rings, the vortex remained in its original position, in some others, that were opened with a larger amplitude, it moved. From this, it is possible to infer the acceleration.
I find the paper very well written, very interesting and about a concrete, timely topic. It presents an exhaustive analysis. Accelerometry with Bose-Einstein condensates is one of the most important applications of quantum physics. It is intriguing that measuring the acceleration, a continuous quantity, is reduced to observing the transition of a vortex from one ring to the other one, which is a discrete event. Thus, I very strongly suggest its publication in SciPost Physics.
Small typo:
Ref. [49] : space is missing between some initials and last names.
Recommendation
Publish (easily meets expectations and criteria for this Journal; among top 50%)

---

## Round 2 · Author Response

We thank the Referees for their reports, and are delighted that they are excited by this work as much as we are. Below, we respond to the requested changes from Referee 1 :

1) The introduction is exhaustive, however I think Dowling's proposal for rotation sensing with a superposition of persistent currents should be included. See: https://arxiv.org/abs/0907.1138

We thank the Referee for highlighting this relevant work. We have included this reference in the next version of the manuscript.

2) In Fig 1, for panels (a) and (b) at t=0, is it worth also stating explicitly that a=0?

The data shown in Fig. 1 were obtained with a finite moving frame acceleration of a=0.01g. Panels (a) and (b) were generated using the imaginary time technique, which yields the stationary solution for a fixed acceleration. These are then compared to the static case (a=0), whose solutions--while not displayed--were discussed in detail in our previous work Phys. Rev. Res. 4, 043171 (2022). Panels (c) and (d) show the differences between the static and accelerated cases.

To improve clarity and eliminate the need for referencing earlier work, we have revised Fig. 1 to include three columns: the non-accelerating case, the accelerating case, and their differences. We believe this presentation is more self-contained and accessible for the reader. We have also clarified how Figure 1 was obtained, referencing the imaginary time technique.

3) In Fig 2, where persistent current oscillations are shown, is there a best time within the oscillation for V0 to begin the ramp to 0, so that the vortex is transferred? If the ramp starts at the wrong time, will the vortex go back to the leading ring?

We thank the Referee for this insightful question. The answer is two-fold. First, the ramp rate has minimal impact on the dynamics, except in the case of an instantaneous ramp. This is because vortex oscillations are suppressed as soon as V<μ, which typically happens shortly after the ramp begins, effectively preventing the scenario raised by the Referee. In the absence of acceleration, this behavior was discussed in our previous work Phys. Rev. Res. 4, 043171 (2022).

However, under acceleration, this warrants further analysis: the critical barrier amplitude becomes acceleration-dependent, altering the timing between initiating the barrier closure and suppressing vortex oscillations. This can be addressed by calibrating the system such that the initial barrier height is as close to V/μ=1 as possible. Moreover, since the oscillation period is independent of acceleration, it is possible to pre-determine an optimal time to begin the ramp.

In response to this question, we have added the following paragraph to our manuscript at the end of section 2.3:

"In Fig. 2, the time at which the barrier ramp-down sequence is initiated affects the final position of the vortex. In the static case, it is known that the oscillations halt as soon as V0 becomes smaller than μ2D. As we will discuss later in the damped regime [Fig. 4], the critical barrier amplitude required to suppress oscillations may be even lower under acceleration. If the goal is to control the final state of the vortex, then this can be achieved by calibrating the maximum barrier amplitude to be as close to the chemical potential as possible. Furthermore, since the oscillation period is independent of acceleration, the optimal barrier closing time can be predetermined."

Best wishes,

Thomas Bland on behalf of all authors

---

## Round 2 · List of Changes

We have made the following changes to the manuscript:

1) Replaced Figure 1 with a new version that shows the static (a=0) case for comparison. The caption and main text has been updated reflecting this change. 2) Added a new discussion on the importance of barrier removal time on configuring the final vortex position. 3) Updated Ref. [49] per Referee 2's request. 4) Updated the final sentence to the abstract, highlighting our proof-of-concept work towards designing the atomtronic accelerometer. 5) Fixed other minor typographical errors in the references and updated DOI links.

---

## Editorial Decision

published